# Information Guided Regularization for Fine-tuning Language Models

**Mandar Sharma**[1], **Nikhil Muralidhar**[2], **Shengzhe Xu**[1]
**Raquib Bin Yousuf**[1] **& Naren Ramakrishnan**[1]
[1]Department of Computer Science, Virginia Tech
[2]Department of Computer Science, Stevens Institute of Technology
mandarsharma@vt.edu

## Abstract

The pretraining-fine-tuning paradigm has been the de facto strategy for transfer learning in modern language modeling. With the understanding that task adaptation in LMs is often a function of parameters shared across tasks, we argue that a more surgical approach to regularization needs to exist for smoother transfer learning. Towards this end, we investigate how the pretraining loss landscape is affected by these task-sensitive parameters through an information-theoretic lens. We then leverage the findings from our investigations to devise a novel approach to dropout for improved model regularization and better downstream generalization. This approach, named *guided dropout*, is both *task & architecture agnostic* and adds *no computational overhead* to the fine-tuning process. Through empirical evaluations, we showcase that our approach to regularization yields consistently better performance, even in scenarios of data paucity, compared to standardized baselines. Our codebase for reproducibilty is hosted here[1].

## 1 Introduction

Although highly-parameterized LMs have recently demonstrated incredible few-shot and zero-shot capabilities (OpenAI, 2023; Touvron et al., 2023), transfer learning via fine-tuning still remains crucial for building task-specific models (Sharma et al., 2022; Min et al., 2023). As such, mechanisms to improve the efficiency of fine-tuning parameter-heavy LMs have garnered much attention leading to frameworks like LoRA (Hu et al., 2021). However, the aspect of transfer learning that doesn't receive as much attention is regularization. As over-parameterized models are made to adapt to niche tasks with few data points, effective regularization can go a long way into building models that generalize better to downstream tasks. Specially in light of recent findings that LMs tend to share parameters depending on the specific downstream task (Sharma et al., 2023), we argue against regularizers that treat all LM parameters the same - *all neurons are equal but some neurons are more equal than others*. Here, we empirically study the effects that LM parameters have on their loss geometry and propose a more surgical and effective approach to regularization.

As our proposed approach is independent of the target task, it simply involves appending a few lines of pre-determined code to the LM training loop to obtain a slight performance improvement on *virtually any downstream task*. This could have significant implications as foundational models are often used for a colossal number of downstream applications - the bert-base-uncased model has been downloaded more than 1.5 billion times on HuggingFace (Wolf, 2024). Concisely, we offer the following contributions:

- We study the effects that task-sensitive parameters have on the LM loss landscape through their visual geometry.
- Through our findings, we propose a novel *information guided* approach to L2 regularization that its both task & architecture agnostic and adds no computational overhead to the fine-tuning process.

---

[1]https://github.com/Mandar-Sharma/Guided-Dropout

- We empirically showcase that our approach to guided dropout yields consistently better performance compared to standardized baselines, specially in scenarios of data paucity.
- Alongside, we also showcase that a reliable estimate of the model information can be obtained in a cost-effective manner simply through a small sub-sample of the training corpus.

## 2 Theoretical Foundations

The theoretical properties underpinning the mini-batch based stochastic gradient descent (SGD) has received thorough exploration from the scientific community (Ge et al., 2015; Hardt et al., 2016; Bottou et al., 2018). Specially relevant to us is the investigation of how loss converges through the approximation of the eigenvalues of the Hessian (Keskar et al., 2016) and through its visualized geometry (Li et al., 2018; Yao et al., 2020). With Fisher information as a proxy to the Hessian (§2.1), we investigate how this computationally cheaper first-order metric can lend us an understanding of how the LM loss landscape geometry is affected by its task-specific parameters (§2.2). Further, we leverage these insights to devise an *information-guided* approach to L2 regularization via dropout (Wan et al., 2013) for transfer learning (§2.3).

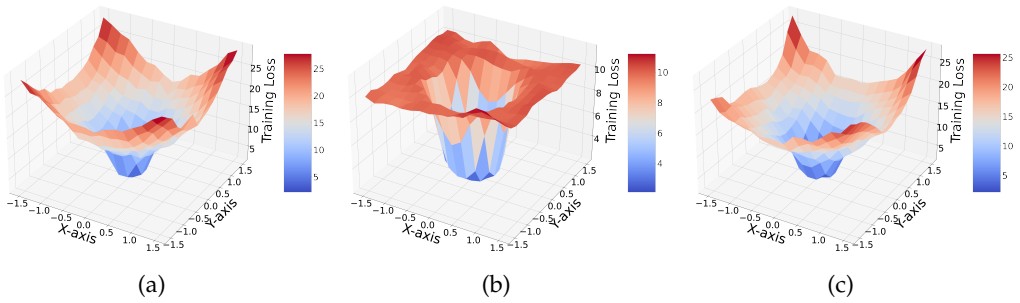

|     |     |     |
| --- | --- | --- |
| (a) | (b) | (c) |

Figure 1: The 3D loss landscape geometry of the pretrained $BERT_{BASE}$ model when: (a) all model parameters are perturbed, (b) 50% of the model parameters with the highest Fisher scores are perturbed, and (c) 50% of the model parameters with the lowest Fisher scores are perturbed. Plot (b) resembles the typical *sharp* minimizers that generalize poorly compared to the *wide* minimizers of base plot (a) and control plot (c).

### 2.1 A Tale of Two Matrices: The Fisher & the Hessian

**Observation 1:** At $\theta \rightarrow \theta_o$, the Fisher information approximates the Hessian and thus can be used to understand the training dynamics and loss landscapes of LMs.

For an LM parameterized by $\theta$, the Fisher score function $s(\theta)$ represents the gradient of the log-likelihood function $\log p(x|\theta)$. The Fisher information $I(\theta)$ is understood as the covariance matrix $K$ of the score function (1). Characteristically, the Fisher score evaluated at the true value of the model parameter $\theta_o$ is 0, giving us (2). However, as this requires computing the expectation over the distribution $\theta$, which is often intractable, the Empirical Fisher (3) is computed given a dataset $x \in X$ with $N$ samples.

$$I(\theta) = K_{s(\theta)} = \mathbb{E}[(s(\theta) - \mathbb{E}[s(\theta_o)])(s(\theta) - \mathbb{E}[s(\theta_o)])^T] \tag{1}$$

$$= \mathbb{E}[s(\theta)s(\theta)^T] \tag{2}$$

$$\equiv \frac{1}{N} \sum_{i=1}^{N} \frac{\partial \log p(x|\theta)}{\partial \theta} \frac{\partial \log p(x|\theta)}{\partial \theta^T} \tag{3}$$

Similarly, the Hessian for the LM characterized by $\theta$ can be written as:

$$H = \frac{1}{N} \sum_{i=1}^{N} \frac{\partial \log p(x|\theta)}{\partial \theta} \frac{\partial \log p(x|\theta)}{\partial \theta^T} - \frac{1}{p(x|\theta)} \frac{\partial^2 p(x|\theta)}{\partial \theta \partial \theta^T} \qquad (4)$$

Jastrzebski et al. (2017) note that the second term in (4) is negligible in the case when the conditional probability distribution of the model coincides with the conditional distribution of the training data. Mathematically, this occurs when the model parameters of the LM $\theta$ is close to the optimum such that $\theta \to \theta_o$. Thus, for a converged pretrained LM, the covariance represented by the Empirical Fisher (3) is approximately the Hessian.

| | Top 5% of Parameters | | | | Top 1% of Parameters | | | |
| | $I_N * I_S$ | | $DKL(I_N, I_S)$ | | $I_N * I_S$ | | $DKL(I_N, I_S)$ | |
| Sample | $\mu$ | $\sigma$ | $\mu$ | $\sigma$ | $\mu$ | $\sigma$ | $\mu$ | $\sigma$ |
|---|---|---|---|---|---|---|---|---|
| 33% | 0.9864 | 0.0409 | 0.0055 | 0.0304 | 0.9767 | 0.0825 | 0.0044 | 0.0243 |
| 10% | 0.9428 | 0.1219 | 0.0164 | 0.0451 | 0.9305 | 0.1665 | 0.0159 | 0.0415 |
| 5% | 0.9661 | 0.0788 | 0.0126 | 0.0544 | 0.9547 | 0.1171 | 0.01 | 0.0407 |
| 1% | 0.9233 | 0.1338 | 0.0261 | 0.0923 | 0.9111 | 0.1732 | 0.0212 | 0.0665 |
| 0.10% | 0.7564 | 0.2448 | 0.1149 | 0.2272 | 0.7349 | 0.3035 | 0.1011 | 0.1757 |
| | Top 0.5% of Parameters | | | | Top 0.25% of Parameters | | | |
| | $I_N * I_S$ | | $DKL(I_N, I_S)$ | | $I_N * I_S$ | | $DKL(I_N, I_S)$ | |
| | $\mu$ | $\sigma$ | $\mu$ | $\sigma$ | $\mu$ | $\sigma$ | $\mu$ | $\sigma$ |
| 33% | 0.9643 | 0.1427 | 0.004 | 0.0208 | 0.9399 | 0.3163 | 0.0034 | 0.0169 |
| 10% | 0.9136 | 0.2057 | 0.014 | 0.0348 | 0.8784 | 0.4215 | 0.0112 | 0.0274 |
| 5% | 0.9445 | 0.1597 | 0.0092 | 0.0351 | 0.8998 | 0.4003 | 0.0072 | 0.0271 |
| 1% | 0.8916 | 0.2504 | 0.0196 | 0.0574 | 0.8424 | 0.4822 | 0.017 | 0.048 |
| 0.10% | 0.7208 | 0.3792 | 0.0898 | 0.1599 | 0.2908 | 0.3956 | 0.0612 | 0.1451 |

Table 1: The layer-wise mean $\mu$ and standard deviation $\sigma$ of the cross-correlation ($*$) and the KL-Divergence (DKL) between the Fisher information computed with the entire WikiText-103 dataset ($I_N$) vs that computed with the sub-sampled dataset ($I_S$, where $S << N$) for the base BERT model. As noted in section §2.2, since only a fraction of the parameters contain the highest concentration of Fisher scores, we evaluate the consistently of the Fisher information matrix based on the top $\{5, 1, 0.5, 0.25\}$ percent of the LM parameters. Based on this table, we conclude that reliable estimates of the Fisher information can be obtained through as little as 1-5 % of the training data.

**Observation 2:** For LMs, reliable estimates of the Fisher information can be obtained from a fractional sub-sampled corpus.

While the eigenvalues of the Fisher matrix can certainly be computed effectively in terms of memory and run-time (Liao et al., 2018), we showcase that for LMs, the entire Fisher information matrix $I(\theta)$ can be approximated through a fractional sub-sample of the original training corpus. The pretraining corpus for $BERT_{BASE}$ consists of a combination of the Wikipedia corpus and the book corpus ($\sim 7$ GB) (Devlin et al., 2019). We attain our approximation of the Empirical Fisher $I(\theta)$ for $BERT_{BASE}$ through WikiText-103 (Merity et al., 2016), a much smaller cut ($\sim 4\%$) of BERT's original pretraining corpus. The effective results we present in section §3 are all based on this approximation. We attribute this observation to the nature of the pretraining LMs - the redundancy that exits as a consequence of being trained on large unstructured corpora of free-form text (Dalvi et al., 2020). To further illustrate this point, in Table 1, we approximate $I(\theta)$ with incrementally smaller sub-samples from WikiText-103. We find reliable estimates of $I(\theta)$ even with just 1% to 5% of the samples from WikiText (0.04% - 0.2% of the original pretraining corpus), although the quality of the estimate drops if we sub-sample below 0.01% (0.004 % of the original corpus).

## 2.2 Fisher Information & the LM Loss Landscape

Prior studies have used the Fisher information as a substitute to the Hessian for characterizing the properties of stochastic gradient descent (Liao et al., 2018; Martens, 2020). To attain complementary insights, we perform an empirical study of the loss geometry of LMs as a function of the Fisher information $I(\theta)$. Borrowing the filter-normalization based visualization scheme of Li et al. (2018), we plot the loss landscape of the pretrained $BERT_{BASE}$ (Devlin et al., 2019) as $f(\alpha, \beta)$ (5) where the cross-entropy loss $L$ is computed as the converged model parameters $\theta_o$ are perturbed with directional vectors $\delta$ and $\eta$ scaled by $\alpha$ and $\beta$ respectively. The loss landscape for the $BERT_{BASE}$ is depicted in Figure 1 (a).

$$f(\alpha, \beta) = L(\theta_o + \alpha\delta + \beta\eta) \tag{5}$$

**Observation 3:** For LMs, an exceedingly tiny fraction of the model parameters have significantly high Fisher scores and they have a pronounced effect on the loss geometry.

LMs are known to have task-specific peculiarities in the parametric distribution of their Fisher information (Sharma et al., 2023). We observe similar behaviors where only an exceedingly tiny fraction of the model parameters have significantly high Fisher scores (see Figure 3 a). To observe the effects of these parameters on the training loss geometry, we selectively perturb half of the parameters of $\theta_o$ that lie on the *higher* end of the Fisher score spectrum (see Figure 1 b). The loss geometry so formed resembles a typical *sharp* minimizer. These have been known to lead to poor generalization (Hochreiter & Schmidhuber, 1997; Keskar et al., 2016; Jastrzebski et al., 2017). On the other hand, the loss geometry of the BERT model with the entirety of $\theta_o$ perturbed (Figure 1 a) resembles that of a *wide* minimizer that are known to generalize better. To act as a control experiment to strengthen findings from Figures 1 a & b, we plot the loss geometry of the same model but this time perturbing those half of $\theta_o$ that lie on the *lower* end of the Fisher score spectrum (see Figure 1 c). This control plot does not deviate from the wide shape of the loss geometry of the $BERT_{BASE}$ model in Figure 1 a. Thus, the half of $BERT_{BASE}$ parameters with higher Fisher scores contribute drastically more to the loss geometry than the half with lower Fisher scores.

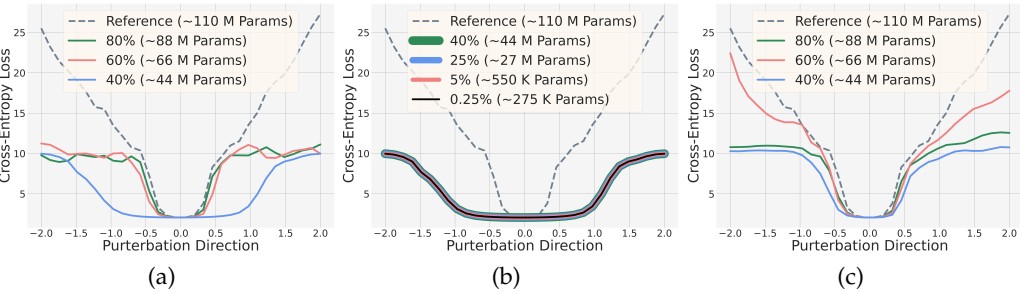

Figure 2: The loss landscape geometry of the pretrained $BERT_{BASE}$ model when: (a) & (b) $n$ % of parameters with the highest Fisher scores are perturbed, (c) $n$ % of randomly chosen parameters are perturbed. Plot (a) showcases worsening minimas as we narrow down on the parameters deemed essential by the Fisher score. Plot (b) showcases that the worsening stops and the landscape remains consisten as the we take $\leq 40\%$ of the highest scoring parameters. Plot (c) acts as the control plot.

Further, incrementally focusing the model perturbation on those selective LM parameters that hold the highest Fisher scores leads to an incremental degradation in the loss landscape (see Figure 2 a). Here, as we perturb fewer and fewer parameters while focusing on the ones with highest Fisher scores (from 80% of the parameters with the highest Fisher scores all the way to 0.25% of the parameters with the highest scores), we see the loss landscape incrementally converging to flatter regions in the neighborhood of the minimum, again leading to poor generalization (Hochreiter & Schmidhuber, 1997; Keskar et al., 2016; Jastrzebski et al., 2017). It is interesting to note that as we go below 40%, the effect of

parameter perturbation on the loss landscape seems to remain consistent. This implies that the 220,000 (0.25%) out of 160,000,000 BERT parameters are so vital to the model convergence that perturbing them produces as pronounced of an effect as perturbing 44,000,000 (40%) of the model parameters. Similar to the 3D loss geometries, to further solidify our observations, we introduce a control plot (see Figure 2 c). In this plot, there is no consistent signal of worsening loss geometry as we perturb *randomly chosen* 80%, 60%, 40% of the model parameters.

## 2.3 Fisher Information & Guided Regularization

**Observation 4:** For transformers, a minority of the layers hold a significant concentration of Fisher sensitive parameters. This observation is used to devise a surgical approach to regularization.

Shifting our attention from the atomic parameter-wise view to an aggregate layer-wise view, we yet again observe similar findings: a minority of the transformer layers hold a significant concentration of Fisher sensitive parameters (see Figure 3b, see Appendix §A.2 for individual non-normalized plots). Based on this observation and our findings from §2.2, we devise an approach to L2 regularization for transfer learning which we term as *information-guided regularization*. Our idea is rather straightforward: implement L2 regularization such that the sway from current model convergence is minimal. In other words, during transfer learning, focus the regularization more on the LM layers deemed less important by the Fisher information and relax regularization on those layers deemed as vital by the same. Based on conventional popularity, we choose dropout (Srivastava et al., 2014) as our L2 implementation. For simplicity, our approach to regularization is algorithmically illustrated in appendix §A.4.

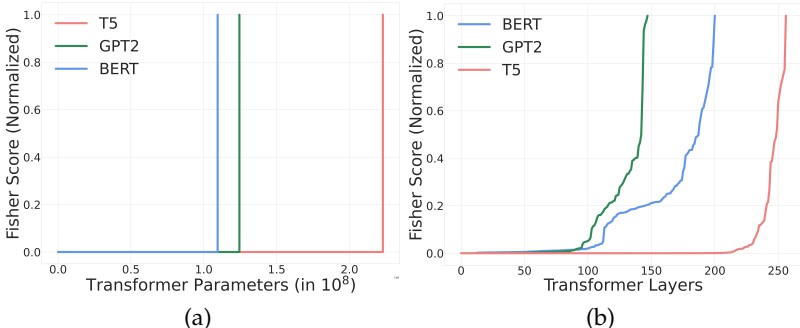

(a)                                            (b)

Figure 3: The sorted Fisher scores (normalized) for BERT, GPT2, and T5 based their (a) model parameters and (b) model layers (aggregated) - see appendix §A.1 and §A.2 for the raw (non-normalized) plots. From (a) we observe that a fraction of LM parameters are often attributed the highest Fisher scores. Similarly, (b) provides an aggregate view showcasing how a minority of the LM layers hold the majority of training information. Please see appendix §A.3 for the unsorted layer-wise Fisher score distributions of these models.

Assuming a LM with $l \in \{1, ..., |L|\}$ layers such that the output of a given layer $l$ with weights $W^{(l)}$ and biases $b^{(l)}$ can be represented as:

$$y^{(l)} = f(W^{(l)}y^{(l-1)} + b^{(l)}) \tag{6}$$

Here, $f$ is the activation function. If layer $l$ is followed by a standard dropout layer in the LM network, the feed-forward operation becomes:

$$r^{(l)} \sim \text{Bernoulli}(p) \tag{7}$$

$$\tilde{y}^{(l)} = r^{(l)} * y^{(l)} \tag{8}$$

Conventional fine-tuning with dropout has $p$ as a constant. Given that we have an ordered collection of layers $S = \{I(l_1) < I(l_2) < ... < I(l_{|L|})\}$ sorted based on their Fisher scores $I(l)$, we propose an incremental setting of layer-wise $p^{(l)}$ such that the masking of $y^{(l)}$ through $Bernoulli(p^{(l)})$ follows a *layer-wise masking schedule*. We define our Bernoulli sampling probability $p^{(l_i)}$ for the layer $l$ at the $i^{th}$ position in $S$ as:

$$p^{(l_i)} = \sum_{j=1}^{i} \Delta s \tag{9}$$

$$r^{(l_i)} \sim Bernoulli(p^{(l_i)}) \tag{10}$$

where $\Delta s = \frac{P_{upper} - P_{lower}}{n_d}$, the step size followed by the *linear increment schedule*. Here, $P_{upper} \in (0, 1)$ and $P_{lower} \in (0, 1)$ are constants that represent the upper and lower bounds for the value that $p^{(l_i)}$ can take and $n_d$ represents the number of dropout layers in the LM architecture. Our approach amounts to sampling sub-networks from a larger network where the guidance imposes a sampling bias towards retaining highly informative neurons in the sub-network. This guided sampling bias leads to improved generalization by learning through a better behaved loss landscape.

The determination of our layer-wise masking schedule is a one-time task. After $p^{(l_i)}$ is determined for a model, the same schedule yields improvements in generalization for all downstream tasks for the model (as we demonstrate in section §3). Thus, not restricted by specific model architectures nor specific downstream tasks, our guided dropout is both *task and architecture agnostic*. Further, as the the layer-wise dropout probabilities $p^{(l_i)}$ are predecided ahead of fine-tuning, guided dropout *adds no computational overhead* to the fine-tuning process. The $p^{(l_i)}$ implementation used for this study is linear, however, users are free to utilize their own non-linear schedules. We empirically evaluate the efficacy of our approach in the section that follows.

## 3 Fine-tuning LMs with Information Guided Regularization

Phang et al. (2018) state that fine-tuning $BERT_{LARGE}$ sometimes fails when a target dataset has fewer than 10,000 training instances. This was specially apparent on a subset of the GLUE dataset (Wang et al., 2018) that have relatively low data samples - CoLA, MRPC, RTE, and STS-B. Intuitively, this would make sense as an overparameterized model would fail to converge on a small dataset. Lee et al. (2019) evaluate their regularization technique by fine-tuning $BERT_{LARGE}$ on these 4 tasks and showcase a high rate of failure among random restarts[2] for standard dropout - specially for the CoLA task where 7 out 20 of their runs fail to converge. In our attempts to replicate this, using the exact same specifications and the original $BERT_{LARGE}$ model, we found that all 30/30 of our random restarts for CoLA converged to a respectable Matthew's correlation score (see appendix §A.5 for the full table). The only differentiating factor between our experiment and theirs was the batch-size, our 8 vs their 32. There have been numerous investigations into how large batch sizes lead to generalization failures (Keskar et al., 2016; Jastrzebski et al., 2017).

However, we do borrow the idea to *simulate* model performance under data paucity. Instead of having an over-parameterized model $BERT_{LARGE}$ evaluated on smaller GLUE datasets, we evaluate $BERT_{BASE}$ on progressively smaller cuts of those same four datasets - CoLA, MRPC, RTE, and STS-B. We progressively shrink the size of the training dataset from 100% to 10% and evaluate our regularizer with standardized baselines all the way through. To establish consistency, we report results as averages across 5 random restart runs along with the corresponding inter-quartile ranges of the runs. Figure 4 showcases the perfor-

---

[2]Based on Lee et al. (2019), random restarts refer to using a different train-test split of the dataset for each fine-tuning run.

mance of guided dropout (our regularizer) against two popular contemporary choices for regularization for the 4 GLUE tasks.

**Baselines:** The standard dropout regularizer (Srivastava et al., 2014) follows the masking of the input vector as shown in equation (6). Gaussian dropout (Wang & Manning, 2013) follows a similar idea where instead of zero-ing out parameters in the input vector, it injects Gaussian noise into the vector. The noise $\epsilon$ draws from the Gaussian distribution $\mathcal{N}(1, \alpha)$ where $\alpha = \frac{p}{1-p}$ with $p$ as the dropout probability. We choose $p = 10\%$ as Lee et al. (2019) report Dropout(0.1) to have be optimal for all GLUE tasks.

**Experimental Setup:** For fair comparisons to the baselines, we use the same training hyperparameters as those recommended by Devlin et al. (2019): an adam optimizer with learning rate = $2 \times 10^{-5}$, $\beta_1 = 0.9$, $\beta_2 = 0.999$, learning rate warm-up over the first 10% of steps and a linear decay of the learning rate after the warm-up steps. For each GLUE task, the exact train-test split is decided by a random seed and the output layer of $BERT_{BASE}$ for the task is randomly initialized from a distribution $\mathcal{N}(0, 0.02^2)$. The fine-tuning is performed for 3 epochs for all tasks and models and the results are evaluated on the development set.

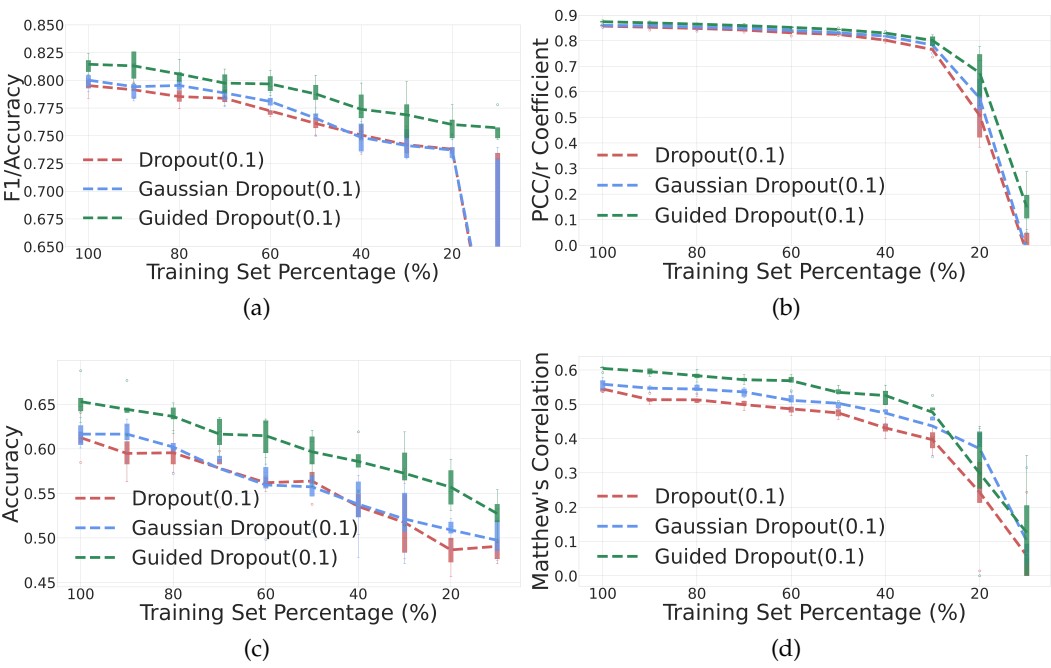

Figure 4: The performance of Guided dropout (our regularizer) vs standardized baselines on fine-tuning $BERT_{BASE}$ on decreasing cuts of the training datasets for (a) MRPC (b) STS-B (c) RTE and (d) CoLA. Each data point in the grid is an average across 5 random restarts, as shown by the boxplots. Guided dropout yields consistently better results for all the tasks, specially under data paucity.

**GLUE Tasks:** Although we evaluate the performance of our model vs the baselines under simulated data-paucity for the select 4 GLUE tasks (following Phang et al. (2018) and Lee et al. (2019), Figure 4), we still perform the standard evaluation our models with respect to the baselines for all 9 tasks in the GLUE benchmark (see Table 2):

- Single Sentence Tasks: CoLA (the Corpus of Linguistic Acceptability) (Warstadt et al., 2019) for grammatical fidelity, SST-2 (the Stanford Sentiment Treebank) (Socher et al., 2013) for sentiment prediction.
- Similarity and Paraphrase Tasks: MRPC (the Microsoft Research Paraphrase Corpus) (Dolan & Brockett, 2005), QQP (the Quora Question Pairs), and STS-B (the Semantic Textual Similarity Benchmark) (Cer et al., 2017) for semantic equivalence.

| Regularizer | CoLA | STS-B | MNLI | MNLI (MM) | MRPC |
|---|---|---|---|---|---|
| Dropout (10%) | $0.5617_{0.5446}$ | $0.8605_{0.8572}$ | $84.52_{84.37}$ | $84.95_{84.47}$ | $80.45_{79.52}$ |
| Gaussian Dropout (10%) | $0.5782_{0.5585}$ | $0.8630_{0.8599}$ | $84.78_{84.26}$ | $\mathbf{85.31}_{84.84}$ | $81.11_{80.01}$ |
| Guided Dropout (10%) | $\mathbf{0.6096}_{0.6045}$ | $\mathbf{0.8813}_{0.8743}$ | $\mathbf{85.11}_{84.42}$ | $85.14_{84.81}$ | $\mathbf{82.41}_{81.44}$ |

| Regularizer | QNLI | QQP | RTE | SST-2 | WNLI |
|---|---|---|---|---|---|
| Dropout (10%) | $89.59_{89.50}$ | $89.53_{89.47}$ | $64.80_{62.92}$ | $94.98_{94.93}$ | $45.07_{43.40}$ |
| Gaussian Dropout (10%) | $89.08_{88.90}$ | $89.45_{89.39}$ | $66.78_{64.25}$ | $95.10_{94.97}$ | $44.36_{43.66}$ |
| Guided Dropout (10%) | $\mathbf{89.71}_{89.64}$ | $\mathbf{89.62}_{89.58}$ | $\mathbf{68.77}_{63.03}$ | $\mathbf{95.23}_{95.23}$ | $\mathbf{50.70}_{45.42}$ |

Table 2: The performance of guided dropout (our regularizer) vs standard and Gaussian dropouts at $p = 10\%$ on GLUE tasks. For consistency, the numbers are presented as $Max_{Mean}$ of results across 5 random restart runs. For CoLA, the metric presented is the Matthew's correlation coefficient and that for STS-B is an average of the Pearson's and Spearman-r coefficient. For all other tasks, the figures presented are averages of the accuracy and f1 scores. Our regularizer outperforms both baselines on 9/10 tasks.

- Inference Tasks: MNLI (the Multi-Genre Natural Language Inference Corpus) (Williams et al., 2018) and RTE (Recognizing Textual Entailment) for textual entailment, QNLI (the Stanford Question Answering Dataset) (Rajpurkar et al., 2016) for Q&A, and WNLI (the Winograd Schema Challenge) (Levesque et al., 2012) for pronoun referent selection.

## 4 Conclusion

Fisher information is a useful tool for characterizing the properties of stochastic gradient descent (Liao et al., 2018; Martens, 2020). We extend its implementation to further understand the training dynamics of LMs through the visual geometry of their loss landscape. Upon observing the pronounced effect of a select few sets of parameters on the LM loss geometry, we devised a surgical approach to L2 regularization that dissuades the sway from optimal model convergence, leading to better generalization in downstream tasks. We term this regularization as *information-guided regularization*. Applied to dropout, this regularization can be comprehended as sampling sub-networks from a larger network where the guidance imposes a sampling bias towards retaining highly informative neurons in the sub-network. This guided sampling bias leads to improved generalization by learning through a better behaved loss landscape. Although surgical, each pretrained model requires this estimate to be computed only once - the same estimate is then applied when fine-tuning to any downstream task. Thus, making our approach both *task & model agnostic* with *no computational overhead* added to the fine-tuning process. We experimentally validate the effectiveness of our regularizer and showcase its prowess, specially in scenarios of data paucity. We also showcase how for LMs, the Fisher information can be frugally approximated with a tiny fraction of the training corpus.

## 5 Limitations

In our study, we extend the use of Fisher information to further understand the training dynamics of LMs through the visual geometry of their loss landscape. From our findings we devise a regularization technique which we showcase to be effective for a multitude of downstream tasks. Due to the exhaustive nature of our experiments, replicating this study for multiple variants of transformer-based LMs was not viable given our in-house computational resources, thus we chose the most popular LM - BERT (Wolf et al., 2019) for our empirical evaluations. However, in section §2, we showcase that the theoretical motivations that make this applicable for the encoder-only BERT model (Devlin et al., 2019) also holds true for the decoder-only GPT2 model (Radford et al.) and the encoder-decoder

based model T5 (Raffel et al., 2020). Thus, we believe our regularizer to be universally applicable.

## 6 Ethics Statement

While many LLM applications are fraught with ethical concerns, we can ascertain that our study does not bring forth additional complications. The datasets we use in this study are established benchmark datasets from publicly accessible websites and do not contain any personally identifiable information. Our analyses does not constitute human subjects and thus is not within the purview of the IRB. A silver lining from our work is that, by adopting regularization techniques as proposed here, we can achieve better downstream generalization without added computational costs.

## 7 Acknowledgements

This work is supported in part by US National Science Foundation grant IIS-2312794.

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

# A Appendix

## A.1 LM Parameters based on their Fisher Scores - *Sorted*

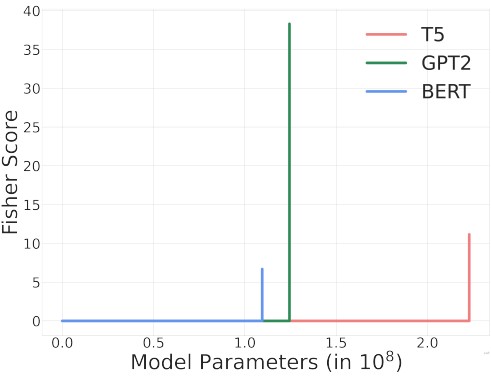

Figure 5: When LM parameters are sorted based on their increasing Fisher scores, we observe a that only an exceedingly tiny fraction of the model parameters have significantly high scores. This observation is consistent for all popular transformer architectures: encoder only (BERT with 0.001% of parameters having $\geq 0.01$ score), decoder only (GPT2 with 0.02% of parameters having $\geq 0.01$ score), and encoder-decoder models (T5 with 0.0009% of parameters having $\geq 0.01$ score).

## A.2  LM Layers based on their Fisher Scores - *Sorted*

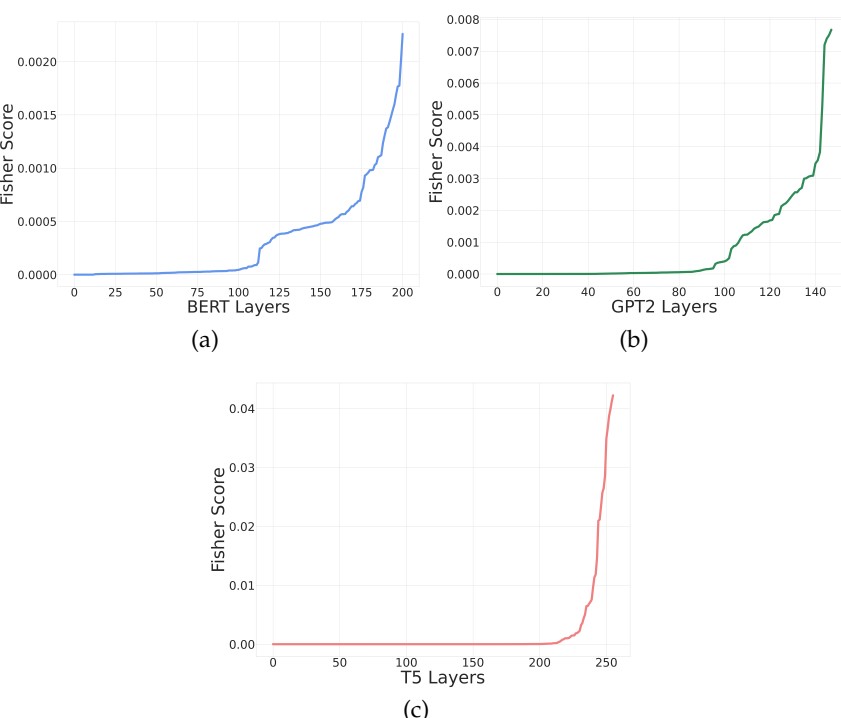

(a)

(b)

(c)

Figure 6: Similar to our observations in §A.1, when the layers of LM are sorted based on the aggregate Fisher score of their respective layer parameters, we observe that certain layers tend to have a concentration of parameters with high Fisher scores. Again, this observation holds true for all popular transformer architectures: (a) encoder only BERT, (b) decoder only GPT2, and (c) encoder-decoder based T5.

## A.3  LM Layers based on their Fisher Scores - *Unsorted*

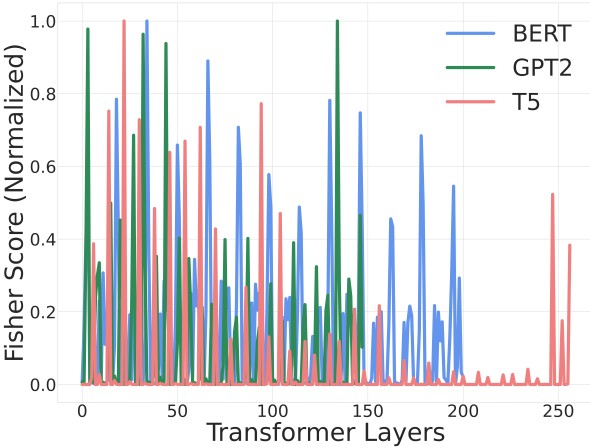

Figure 7: Complementary to figure 6, while only certain layers of the model have significantly high scores, these vital layers are spread throughout the network and not concentrated towards the beginning or end of the network as traditionally believed.

## A.4 Algorithmic Representation of Information Guided Fine-tuning

---

### Information-Guided Regularizaiton - The Algorithm

---

1. For pre-training corpus $C$ with $N$ samples used to train a foundational model $\Theta$, let us subsample $n = 1\%$ of $N$ samples (Page 3, Table 1)

2. With these $n$ samples, we estimate the empirical Fisher $I(\Theta)$ (Page 2, Equation 3) for $\Theta$. Based on $I(\Theta)$, we compute the layer-wise aggregate Fisher scores

3. If $\Theta$ = BERT$_{BASE}$, there are 37 dropout layers in its architecture. These dropout layers are arranged in the descending order of the Fisher scores of the encoder layers that precede them (*dropout_layer_ids* in the code snippet below).

4. With P$_{Upper}$ = 10% and P$_{Lower}$ = 0%, we linearly decrease the dropout rate for layers proportional to the increasing Fisher scores (Page 6, Equations 9 & 10) Thus, less important layers are highly regularized and more important layers are lightly regularized. The programmatic snippet appears as the following:

```
1 def assign_dropout(dropout_layer_ids -> List[int], p_upper -> float, p_lower -> float):
2      #Design a linearly decreasing dropout schedule
3      dropout_probs = range(p_upper, p_lower, step_size)
4      assert len(dropout_probs) == len(dropout_layer_ids)
5      #Assign dropout probabilities based on increasing Fisher score
6      for layer_id, layer_params in enumerate(model.modules()):
7          #If module the layer is a dropout layer
8          if layer_id in dropout_layer_ids:
9              #Find its Fisher-based ranking by its index in dropout_layer_ids
10             idx = dropout_layer_ids.index(layer_id)
11             #Assign the respective dropout probability from the schedule dropout_probs
12             layer_params.dropout_probability = dropout_probs[idx]
```

4. Fine-tune the model to *any* downstream task after modifying the dropout probabilities. Please note that the above code block is *not* dependent on the target task. if the foundational model $\Theta$ remains the same, the same dropout scheme will improve performance on any and all target tasks (Figure 4 and Table 2)

---

## A.5 Fine-tuning BERT Large on CoLA

| Restart Seed | Matthews Correlation | Restart Seed | Matthews Correlation |
|:---:|:---:|:---:|:---:|
| 1 | 0.6357 | 16 | 0.6074 |
| 2 | 0.6164 | 17 | 0.6657 |
| 3 | 0.6127 | 18 | 0.6186 |
| 4 | 0.6412 | 19 | 0.6437 |
| 5 | 0.6060 | 20 | 0.6280 |
| 6 | 0.6730 | 21 | 0.6266 |
| 7 | 0.6482 | 22 | 0.5982 |
| 8 | 0.6206 | 23 | 0.6567 |
| 9 | 0.6475 | 24 | 0.6205 |
| 10 | 0.6555 | 25 | 0.6390 |
| 11 | 0.6357 | 26 | 0.6378 |
| 12 | 0.6348 | 27 | 0.6461 |
| 13 | 0.6710 | 28 | 0.6275 |
| 14 | 0.6470 | 29 | 0.6759 |
| 15 | 0.6396 | 30 | 0.6394 |

Table 3: The seed values and the respective Matthew's Correlation score for 30 random restart runs of $BERT_{LARGE}$ fine-tuned on CoLA.

