# OpenReview forum: "Information Guided Regularization for Fine-tuning Language Models"
_colmweb.org/COLM/2024/Conference — COLM_

### Official Review · Reviewer_bMGQ · 2024-04-21

**Rating:** 5
**Confidence:** 4
**Ethics Flag:** 1

**Summary:**

This paper examines the relationship between task-sensitive parameters and the loss landscape during pre-training. It investigates dropouts and introduces a new variant called 'guided dropout.' The experiments conducted demonstrate that the proposed guided dropout consistently yields better performance.
The clarity of the writing is fair (but with some technical details missing), however execution quality, originality of the work can be improved (see reasons to reject).

**Questions To Authors:**

- How are P_upper and P_lower determined, if the masking follows a schedule, what’s the typical schedule? An analysis would be nice.

- Table 1: Prior work already uses a fraction of training data for estimating FIM. This experimental finding only holds when computing FIM with respect to a subset of parameters. This should be clarified further in Observation 2 in writing.  While this analysis is interesting, would you please clarify its connection with the layer-wise masking dropout discussed later in the paper?

- How does the proposed method work on other type of transformers (other than BERT)?


-----
Updated my score to reflect the opinion change after rebuttal.

**Reasons To Accept:**

- The author proposed an interesting training method by sampling layers for masking out.
- The proposed guided dropout seems simple to implement and is effective with the BERT model.
- The cross-correlation analysis on the effects of training data and sparsity for FIM calculation provides valuable insights, challenging common assumptions in this area.

**Reasons To Reject:**

- While containing interesting experiments, the connection between FIM, sparsity, and the proposed algorithm is unclear. It would be better if the author provided an algorithm block. Currently, it is unclear how Fisher information is used as part of the algorithm.
- The paper ignored prior literature that studied sparsity, fisher information, and transfer learning with no related work section. This leads to potential insufficient baselines, given a few examples (all can serve as baselines in fact…):

  - Sung, Yi-Lin, Varun Nair and Colin Raffel. “Training Neural Networks with Fixed Sparse Masks.”   – using FIM to determine a submask for training transformers, experimented with GLUE tasks and BERT as the backbone.
  - Liu, Chen Cecilia, Jonas Pfeiffer, Ivan Vulic and Iryna Gurevych. “FUN with Fisher: Improving Generalization of Adapter-Based Cross-lingual Transfer with Scheduled Unfreezing.” – using FIM to determine the training order of adapters for cross-lingual transfers with transformers.
  - Lodha, Abhilasha, Gayatri Belapurkar, Saloni Chalkapurkar, Yuanming Tao, Reshmi Ghosh, Samyadeep Basu, Dmitrii Petrov and Soundararajan Srinivasan. “On Surgical Fine-tuning for Language Encoders.”  - using FIM to determine which layer to train with GLUE tasks and BERT backbone.

There are also non-FIM based varieties in terms of (temporarily) omitting layers for training such as LP-FT, gradual unfreezing or Auto-RGN/Auto-SNR in surgical fine-tuning…

---

> ### Author Rebuttal · Authors · 2024-05-30
>
> Thank you for taking the time to review our manuscript. Your insightful comments help us better our research. Please let us address your concerns below:
>
> 1. We agree and will include this block in the revised manuscript to clarify the implementation of FIM as well as the P_upper & P_lower based schedules - see [algorithm](https://shorturl.at/buIh5).
>
> 2. While we agree that these papers related to FIM & NN training would provide a fruitful background to our readers, they are solving fundamentally different problems:
>
> **Their Problem:** Using parameter-efficient fine-tuning (PEFT) via sparser models, can we *match* standard fine-tuning using the full dataset, by *computing FIM for each target task*?
>
> **Our Problem:** With the architecture unchanged, can we *beat* standard fine-tuning, especially in scenarios of data paucity, by *computing FIM only once for the source task*?
>
> Nevertheless, we did the comparison suggested by the reviewer and *our method indeed beats the latest state-of-the-art in the list (Lodha 2023)  by a significantly large margin*, see [table](https://shorturl.at/j9VVy).
>
> Regarding the other baselines mentioned:
>
> - Liu 2024 use FIM as an analytical tool for their chosen domain of cross-lingual adaptation rather than as a general purpose regularizer, as done here. Further, this paper was published on Arxiv after the COLM submission deadline.
>
> - LP-FT, gradual unfreezing (similar to Liu 2024), and auto-RGN/SNR are PEFT as well. RGN/SNR (similar to Sung 2021 & Lodha 2023) are based on target tasks.
>
> To address your questions:
>
> 1) With a grid-search, we found P_lower = 0% and P_upper = 10% with a linear increment schedule to be the most effective approach. We will add grid-search results to the Appendix if our manuscript is accepted. Thank you for bringing this to our attention.
>
> 2) Consider A to be the FIM with scores for each parameter computed w.r.t the full corpus. Table 1 evaluates A with matrices B_n - FIMs with scores for each parameter computed w.r.t *n* % of the corpus. From observation 3, we know that the information from FIMs we need to pull only pertains to the top *x* % of parameters. Thus, The FIM approximates B_n are not computed w.r.t a subset of parameters but are rather evaluated w.r.t them.
>
> 3) Yes. The central requirement for guided dropout to function is the fact that modern LMs are overparameterized. In section 2.3, we show that this thesis holds true for both GPT2 (decoder based) and T5 (encoder-decoder based) models.

---

> > ### Comment · Reviewer_bMGQ · 2024-06-03
> > **Re authors**
> >
> > Thank you very much for the additional information.
> >
> > *Their Problem vs Our Problem*: their problem don't perform architectural change...
> > Sung et al. can be seen as parameter-wise dropout using FIM.
> > Liu et al. is 2023, not 2024 -->  can be seen as layer-wise progressive dropout using FIM
> > RGN/SNR (Lee et al, 2022 from surgical FT) --> archiving the similar effect by reduce the update of parameters that get's small update --> parameter-wise
> >
> > - It is interesting and important to compare the strength of layer-wise versus parameter-wise method especially FIM are estimated.
> >
> > - Thank you very much for the additional experiment using Lodha 2023.

---

### Official Review · Reviewer_15W3 · 2024-05-10

**Rating:** 6
**Confidence:** 4
**Ethics Flag:** 1

**Summary:**

Based on the relationship between (empirical) Fisher information and (the expected) hessian matrix, as well as the established observation that "sharpness" of Hessian's correlation with generalizability, this paper proposes to measure "how well the pretrained parameters are trained" based on the scale of Fisher score, i.e., parameters with higher Fisher scores are less well pretrained. Built on this conjecture, this paper developed a variant of dropout to apply different strengths of regularization to different layers during the fine-tuning stage. Overall I find this paper interesting and potentially useful in various downstream applications.

**Questions To Authors:**

See the first point in reasons to reject.

**Reasons To Accept:**

1. Simple yet effective method
2. Illustrative analyses
3. Efficient estimation of Fisher scores

**Reasons To Reject:**

1. I think the authors can improve their writing by offering more details. Especially, I am curious about 1) what is the actual cost of Fisher score estimation? 2) which layers are with higher and lower Fisher scores? 3) In Figure 2, we see a big difference between 40&60, what happened there?
2. Since the loss visualization is quite central to the analyses, offering a brief preliminary section would be beneficial (longer and more detailed than the part in 2.2).
3. The paper in my opinion would be strengthen by adding more experiments: 1) adding some experiments on OOD. I wonder whether keeping flatter parameters can benefit OOD generalization, and 2) (less important) adding experiments on larger decoder models. As I understand the compute requirement here, I believe adding more "modern" architectures will better show the effect of guided dropout.

---

> ### Author Rebuttal · Authors · 2024-05-30
>
> Thank you for taking the time to review our manuscript and for your positive remarks. These insightful comments help us better our research. Please allow us to address your concerns and the questions you have raised:
>
> 1. If our manuscript is accepted, we will append the following details to section 2:
>
> - For a network with *|W|* parameters, the cost of estimating Fisher information using *n* samples is *O(n|W|)* gradient computations in the network. We showcase that reliable estimates of Fisher information can be obtained with *n* as tiny as 1% of the pretraining corpus (table 1).
>
>
> - While our current plots showcase sorted layer-wise Fisher scores, we will append figures (see [here](https://shorturl.at/jVyou)) to showcase the raw layer-wise distributions. Interestingly, we note that vital layers are spread throughout the network and not concentrated at the beginning or end as traditionally believed.
>
>
> - We will expand on the details of the loss-landscape computation and visualisation through more elaborate writing and by mathematically extending aspects of equation 5.
>
> 2. Figures 2 a & b serve to show the profound effect that a few vital parameters have on the model's loss geometry. 2(b) shows that perturbing the top 0.25% of parameters has the same effect on the loss geometry as perturbing the top 40%. You raise an important question - where is the tipping point (40% < x < 60%) such that the degradation of the loss landscape turns from gradual to sudden. We believe this warrants an "edge of chaos" style analysis [3] which we aim to do in future work.
>
> 3. OOD generalization across similar tasks as a function of Fisher-based sharper/flatter parameters is an interesting question. There is evidence indicating that Fisher-guided pretraining-finetuning helps generalize better to OOD instances [1]. In line with this, we will evaluate OOD generalization for our method in future work.
>
> 4. We agree that if applied to modern architectures that are even more overparameterized [2], the effects of guided-dropout would be even more pronounced.
> Although we personally do not have the compute to evaluate larger models, we strongly believe our findings should generalize.
>
> References:
>
> 1. Sharma et. al. Learning Non-linguistic Skills without Sacrificing Linguistic Proficiency. ACL Proceedings. 2023.
>
> 2. Gromov et. al. The Unreasonable Ineffectiveness of the Deeper Layers. 2024.
>
> 3. Zhang et. al. Edge of Chaos as a Guiding Principle for Modern Neural Network Training. 2021.

---

> > ### Comment · Reviewer_15W3 · 2024-06-02
> >
> > Thank you for your reply. I think most my concerns are addressed. I'll keep the score while leaning towards acceptance.

---

### Official Review · Reviewer_vAfW · 2024-05-11

**Rating:** 7
**Confidence:** 3
**Ethics Flag:** 1

**Summary:**

This paper proposes a novel approach to regularization in the context of fine-tuning language models (LMs). By utilizing Fisher information to analyze the effects of task-sensitive parameters on the loss landscape of pretrained models, the authors introduce a method called "guided dropout." This technique adjusts dropout rates layer-wise based on the Fisher information of parameters, aiming to minimize the perturbation from optimal convergence and enhance model generalization, especially under conditions of limited data availability. Their empirical results suggest that this approach outperforms traditional regularization methods across various tasks, particularly in data-scarce scenarios.

**Reasons To Accept:**

1. The use of Fisher information to guide dropout regularization provides a novel and theoretically motivated method to improve LM generalization, particularly in fine-tuning phases.

2. The proposed method is versatile, applicable to various LMs (as demonstrated with BERTBASE, GPT2, and T5), and independent of the specific downstream tasks, which enhances its utility across different settings.

3. The comprehensive empirical testing, including comparisons with standard regularization methods across multiple GLUE tasks, robustly supports the effectiveness of the guided dropout approach.

**Reasons To Reject:**

1. While the results are promising, the experiments are limited to a specific subset of models and tasks. Extending these findings to a wider array of models and scenarios would be beneficial to substantiate the universality of the approach.

2. The effectiveness of the approach is heavily reliant on the accurate estimation of Fisher information, which might not always be feasible or reliable, especially with very large models or complex datasets.

3. Despite claims of architecture agnosticism, the actual implementation and effectiveness may vary significantly across different architectures, which was not fully explored beyond a few model types.

---

> ### Author Rebuttal · Authors · 2024-05-30
>
> Thank you for taking the time to review our manuscript and for your positive remarks. These insightful comments help us better our research. Please allow us to address your concerns and the questions you have raised:
>
> 1. We agree that extending our findings to a greater array of models and datasets would further help our cause. With statistics presented for 5 random restarts, showcasing the efficacy of our approach under data sparsity for downstream tasks requires fine-tuning all models with several cuts of the training dataset (from 100% to 10%). Thus, to accommodate both compute and page constraints, we chose the tried and tested GLUE benchmark. However, if our manuscript is accepted, we will utilize the extra page to extend our experimentation to the SuperGLUE benchmark [1] as well as other models.
>
> 2. Yes, the effectiveness of our approach indeed relies on the accurate estimation of the Fisher information. Thus, through observation 2 & table 1, we establish that for a text-based pretraining corpus (central to all LLMs), the Fisher information can be reliably estimated using as little as 1% of the training data (rows 4 & 8 in table 1), making our method tractable even for large complex datasets and models.
>
> 3. The foundational principle behind our approach is the fact that modern LMs are overparameterized. This overparameterization leads to a small subset of model parameters driving the training loss geometry (observation 3, page 4). Following this, our claim for architectural agnosticism mainly stems from two observ​​ions:
>
> - In section 2.3, we show that the same overparameterization that held true for the encoder-only BERT also holds true for the decoder-only GPT2 and the encoder-decoder based T5. Although modern LMs such as GPT3/4, Llama, and Mistral are significantly larger, they operate on the same decoder-only architecture as GPT2.
>
> - Recent work has shown that these larger decoder-only models are as overparameterized (if not more) than their smaller counterparts. Gromov et. al. 2024 [2] showcase that up to 50% of Llama 70B, 40% of Llama 13B, and 30% of Mistral parameters are redundant. Thus, although we personally do not have the compute to evaluate larger models, we strongly believe our findings should generalize.
>
> References:
>
> 1. Wang et. al. SuperGLUE: A Stickier Benchmark for General-Purpose Language Understanding Systems. 2020.
>
> 2. Gromov et. al. The Unreasonable Ineffectiveness of the Deeper Layers. 2024.

---

### Official Review · Reviewer_NchA · 2024-05-14

**Rating:** 6
**Confidence:** 4
**Ethics Flag:** 1

**Summary:**

The paper performs extensive empirical studies on the relationships between Fisher information and loss landscape during LM training.
They propose an adaptive dropout technique based on their empirical findings, which has demonstrated to be effective across multiple downstream tasks and multiple types of LMs.

**Questions To Authors:**

What do you think of its potential generalization to larger decoder-only models?

What do you think of its potential generalization to other architectures, like SSM?

**Reasons To Accept:**

The results are very interesting to know. So I think it will be good for the community to see this paper.

The 4 key observations are very interesting and I enjoy reading about them.

The empirical results verify their hypothesis and demonstrate their proposed methods.

The experiments are solid.

**Reasons To Reject:**

The experiments are somewhat limited.

BERT, T5, and GPT2 are very interesting models, but they are somewhat obsolete.

The authors claim that "Thus, we believe our regularizer to be universally applicable."
But this seems to be over-claim because there are no experiments---please correct me if I am wrong---about new generation of LMs like Llama or Mistral (even the smallest versions).

---

> ### Author Rebuttal · Authors · 2024-05-30
>
> Thank you for taking the time to review our manuscript and for your positive remarks. These insightful comments help us better our research. Please allow us to address your concerns and the questions you have raised:
>
> 1. Models like BERT/GPT2/T5, while superseded by larger models, are still useful to demonstrate the validity of theoretical constructs like ours (see other recent papers like [1,2]). Current limitations in resources prevent us from applying our approach to models like Llama/Mistral. However, as we believe all modern LMs to be overparameterized, our methods should transfer over:
>
> **Architectural Agnosticism:** Overparameterization drives redundancies in the training loss geometry that can be harnessed by intelligent dropout mechanisms like ours.  In section 2.3, we show that the same overparameterization that held true for the encoder-only BERT also holds true for the decoder-only GPT2 and the encoder-decoder based T5. Although modern LMs such as GPT3/4, Llama, and Mistral are significantly larger, they operate on the same decoder-only architecture as GPT2.
>
> **Prevalence of Overparameterization:** Further, recent work has shown that these larger decoder-only models are as overparameterized (if not more) than their smaller counterparts. Gromov et. al. 2024 [3] showcase that up to 50% of Llama 70B, 40% of Llama 13B, and 30% of Mistral parameters are redundant. Thus, although we currently do not have the compute required to evaluate larger models, we strongly believe our findings should generalize.
>
> 2. While SSMs like Mamba [4] have showcased transformer-esque performance with faster inference and better scaling, their application to the language modality still follows the pretrain-finetune paradigm. Thus, our regularizer *should* improve downstream performance. Although gauging the *magnitude* of improvement requires a Fisher-based investigation similar to section 2. If our manuscript is accepted, we will include an investigation about the applicability of our method to non-transformer based architectures.
>
> References:
>
> 1. Sharma et. al. Learning Non-linguistic Skills without Sacrificing Linguistic Proficiency. ACL Proceedings. 2023.
>
> 2. Lodha et. al. On Surgical Fine-tuning for Language Encoders. EMNLP Findings. 2023.
>
> 3. Gromov et. al. The Unreasonable Ineffectiveness of the Deeper Layers. 2024.
>
> 4. Gu et. al. Mamba: Linear-Time Sequence Modeling with Selective State Spaces. 2023.

---

### Decision · Program_Chairs · 2024-07-10

**Decision:**

Accept

**Comment:**

This paper introduces  and "information guided" for regularization through dropout; this is proposed as a regularization technique for fine-tuning language models. The method uses Fisher information as a proxy to the Hessian to infer attributes of the loss landscape and use that to adjust dropout rates layer-wise accordingly. The authors claim improved model generalization, especially in data-scarce scenarios, and demonstrate consistent performance gains across various GLUE tasks.

Strengths:
+ theoretically grounded approach using Fisher information for regularization
+ method applicable to various LM architectures (BERT, GPT2, T5)
+ nice analysis of loss landscape and efficient estimation of Fisher information

Weaknesses (-):
- Limited experimentation on newer, larger language models (e.g., Llama, Mistral)
- Lack of comparison to other regularization techniques beyond standard dropou
- "information theory" link pretty weak. i would suggest to rewrite these claims
- Quality drop in Section 3 compared to earlier sections
- Missing details on hyperparameter sensitivity and implementation
- weak related work section


Reviewer Consensus:
The reviewers generally view the paper positively, acknowledging its potential contribution. Some praise the strength of the first two sections on loss landscape analysis and FIM estimation. However, concerns were raised about the quality of Section 3, limited experimentation with newer models, and the absence of a related work section.

Reviewer bMGQ, initially critical, did recognized some of paper's merits. I am personally leaning towards acceptance, recognizing the paper's contribution and potential value to readers, despite some limitations.

[comments from the PCs] As much as possible, please follow up on the reviewer and AC comments to improve the paper.